# Developing a programme theory of a complex, home-based rehabilitation intervention for recovery after an episode of delirium

Shruti Raghuraman[1]*, Sarah Morgan-Trimmer[2], Rob Anderson[1],
Victoria A. Goodwin[1,3], Linda Clare[1,3], Ellen Richards[4], Alison Bingham[1],
Elizabeth Goodwin[5], Rowan Harwood[6], Annie Hawton[5], Aseel Mahmoud[1],
Sarah Joanna Richardson[7], Jinpil Um[1], Obioha C. Ukoumunne[3], Louise Allan[1,3]

**1** Faculty of Health and Life Sciences, Department of Health and Community Sciences, University of Exeter, Exeter, United Kingdom, **2** School of Primary Care, Population Sciences and Medical Education, University of Southampton, Southampton, United Kingdom, **3** NIHR Applied Research Collaboration South West Peninsula, University of Exeter, Exeter, United Kingdom, **4** Royal Devon University Healthcare NHS Trust, Exeter, United Kingdom, **5** Health Economics Group, University of Exeter Medical School, University of Exeter, Exeter, United Kingdom, **6** School of Health Sciences, University of Nottingham, Queens Medical Centre, Nottingham, United Kingdom, **7** AGE Research Group, NIHR Newcastle Biomedical Research Centre, Newcastle upon Tyne Hospitals NHS Foundation Trust, Cumbria Northumberland Tyne and Wear NHS Foundation Trust and Faculty of Medical Sciences Newcastle University, Newcastle upon Tyne, United Kingdom

* s.raghuraman@exeter.ac.uk

## Abstract

### Background

Delirium in older adults is associated with persistent cognitive and functional decline, increased institutionalisation, and higher mortality. Evidence-based strategies to support recovery after hospitalisation remain limited. This paper outlines the development and refinement of the programme theory underpinning RecoverED, a novel, home-based, multicomponent rehabilitation intervention designed to support post-delirium recovery.

### Methods

We applied a realist-informed, multi-stage process to develop the RecoverED programme theory. This included a rapid realist review, qualitative interviews with older adults, carers, and healthcare professionals, an expert panel workshop, iterative programme theory meetings, and a process evaluation embedded within a single-arm feasibility trial. Data were synthesised into a logic model linking intervention components to hypothesised mechanisms and outcomes.

### Results

The programme theory outlines how cognitive and physical rehabilitation, psycho-social support, health monitoring, lifestyle guidance, and delirium education are

**Data availability statement:** No new primary data were generated as part of this publication. The datasets underpinning the individual studies are available through their respective publications. The minimal dataset underlying the findings reported in this manuscript comprises anonymised qualitative interview excerpts (Phase 2) used to inform the analyses and conclusions presented. These coded interview extracts have been made publicly available via the institutional repository ORE (https://doi.org/10.24378/exe.31161238) with all potentially identifying information removed in accordance with ethical approvals and participant consent. Additional methodological detail required to interpret the data is provided within the paper.

**Funding:** This study is funded by a start-up grant from the University of Exeter and by the National Institute of Health and Care Research Programme Grants for Applied Research Programme (NIHR202338). It is supported by the NIHR Applied Research Collaboration South West Peninsula (PenARC), Exeter NIHR Biomedical Research Centre and the Exeter Clinical Trials Unit. Professor Goodwin is a National Institute for Health and Care Research (NIHR) Senior Investigator. Sarah Joanna Richardson holds an National Institute for Health and Care Research (NIHR) Clinical Lectureship, funded by Health Education England (HEE)/NIHR, and is supported by the NIHR Newcastle Biomedical Research Centre (reference: NIHR203309). The views expressed in this publication are those of the authors and do not necessarily reflect the views of NIHR, NHS or the Department of Health and Social Care. The funders had no role in study design, data collection and analysis, decision to publish, or preparation of the manuscript.

**Competing interests:** The authors have declared that no competing interests exist.

expected to promote recovery. Cognitive activities aim to rebuild executive function and daily independence; physical rehabilitation maintains mobility; psychosocial support reduces anxiety and promotes confidence; health monitoring and lifestyle guidance address comorbidities; and delirium education supports sense-making. Key mechanisms include personalised goal-setting, continuity of professional support, and integration with community services. The process evaluation within the feasibility trial confirmed the relevance and acceptability of these pathways, while suggesting refinements to training, psychosocial strategies, and education delivery.

## Conclusion

This paper provides an empirically grounded framework detailing the development and refinement of a programme theory for post-delirium rehabilitation, illustrating the value of flexible, theory-driven approaches to guiding recovery.

---

## Introduction

Delirium is an acute-onset neuropsychiatric syndrome in hospitalised older adults, with those affected experiencing significantly worse long-term outcomes post-discharge. These include persistent cognitive and functional decline, increased risk of dementia, reduced quality of life, greater mental health burden, higher likelihood of institutionalisation, increased hospital readmissions, and elevated mortality [1]. Currently, optimal strategies to support recovery after hospitalisation for delirium remain unclear. Given the breadth of diverse and interconnected adverse outcomes, a multi-disciplinary, holistic rehabilitation approach is urgently needed to address the syndrome's wide-ranging and lasting effects.

RecoverED (**Recover**y after an **E**pisode of **D**elirium) is a novel, multi-component, complex intervention designed to support recovery from an episode of delirium. It is designed to be delivered in the homes of older people who have been discharged from acute settings. A multidisciplinary team (MDT) consisting of physiotherapists (PTs), occupational therapists (OTs) and rehabilitation support workers (RSWs) plan and deliver the intervention over a course of approximately 12 weeks. The intervention has been developed and evaluated in a multi-stage programme of studies, detailed elsewhere.

This article describes the realist-informed approach used to develop the programme theory underpinning the RecoverED intervention, which guided its design and implementation. We detail the iterative development of the theory and model to inform a novel, complex, home-based rehabilitation intervention for older adults following hospitalisation for delirium. We then present findings from a process evaluation embedded within a single-arm, multi-site feasibility study, aimed at refining and contextualising the programme theory. Finally, we discuss key uncertainties and future directions to support further development and implementation of the RecoverED intervention.

## Developing programme theory for RecoverED

The Medical Research Council (MRC) guidance on developing and evaluating complex interventions highlights the need for theory-driven approaches that account for contextual complexity and the interplay of systems, mechanisms and implementation processes in shaping outcomes [2]. Traditional experimental and quasi-experimental designs have been criticised for oversimplifying causality and neglecting implementation context [3]. In contrast, realist evaluation assumes that context-sensitive mechanisms generate outcomes and seeks to explain how, why, for whom, and under what circumstances interventions work [4,5]. This 'white box' approach allows evaluators to systematically examine the mechanisms and contexts underlying programme effects [4,6,7].

The MRC framework recommends developing a programme theory to articulate key components of the intervention, their interactions, and the interplay between casual mechanisms and contextual factors [2]. Programme theories are useful for linking intervention activities with outcomes and outlining how and why the desired changes are expected to take place. In the RecoverED project, the programme theory was developed prospectively to inform which intervention components might be needed and causally important, as well as to guide understanding of how and why these components could work and be optimally implemented within a specific context [8].

This study first presents a detailed programme theory in the form of a logic model, visually illustrating how the intervention strategies and outcomes are grounded in theory to support subsequent evaluation [9]. Logic models are commonly used in theory-driven evaluations to assess feasibility, clarify goals, identify conceptual gaps, monitor implementation, develop evaluation measures, and support dissemination [9].

We describe the iterative development of the programme theory to guide the design of a novel, complex, home-based rehabilitation intervention for older adults with delirium following discharge from acute care. We then report findings from a process evaluation conducted within a single-arm, multi-site feasibility study, aimed at reflecting upon the programme theory. While initially intended to be refined as realist-informed Context-Mechanism-Outcome configurations, their ultimate form was at a higher level of abstraction, more closely linked to intervention components and activities, and the specific problems of recovery after delirium that they intended to reduce. Finally, we outline key uncertainties and future directions to support the further development and implementation of the RecoverED intervention.

## Methods

Adapting the approach used by Renger et al. [10] and Ebenso et al. [9], a realist evaluation perspective was taken to first explore what contextual factors are relevant and should be prioritised when thinking about community-based delirium rehabilitation for older people. Next, we identified and prioritised key contextual elements, and integrated them into a preliminary version of a programme theory. This process elaborated previously defined domains or essential components to provide a logical sequence of how intervention activities were expected to operate within certain contexts to effect change and result in improved outcomes. The iterative, theory-driven, multi-stage process of data collection, analysis, and evaluation is depicted in Fig 1.

## Ethics statement

This paper reports a secondary synthesis of findings from several previously published studies that collectively informed the iterative development of the programme theory. Each underlying study (where relevant) obtained ethical approval from the relevant institutional research ethics committee and followed approved procedures for informed consent, capacity assessment, and data management, as described in those respective publications. No new data collection, participant involvement, or additional ethics review was required for the current publication.

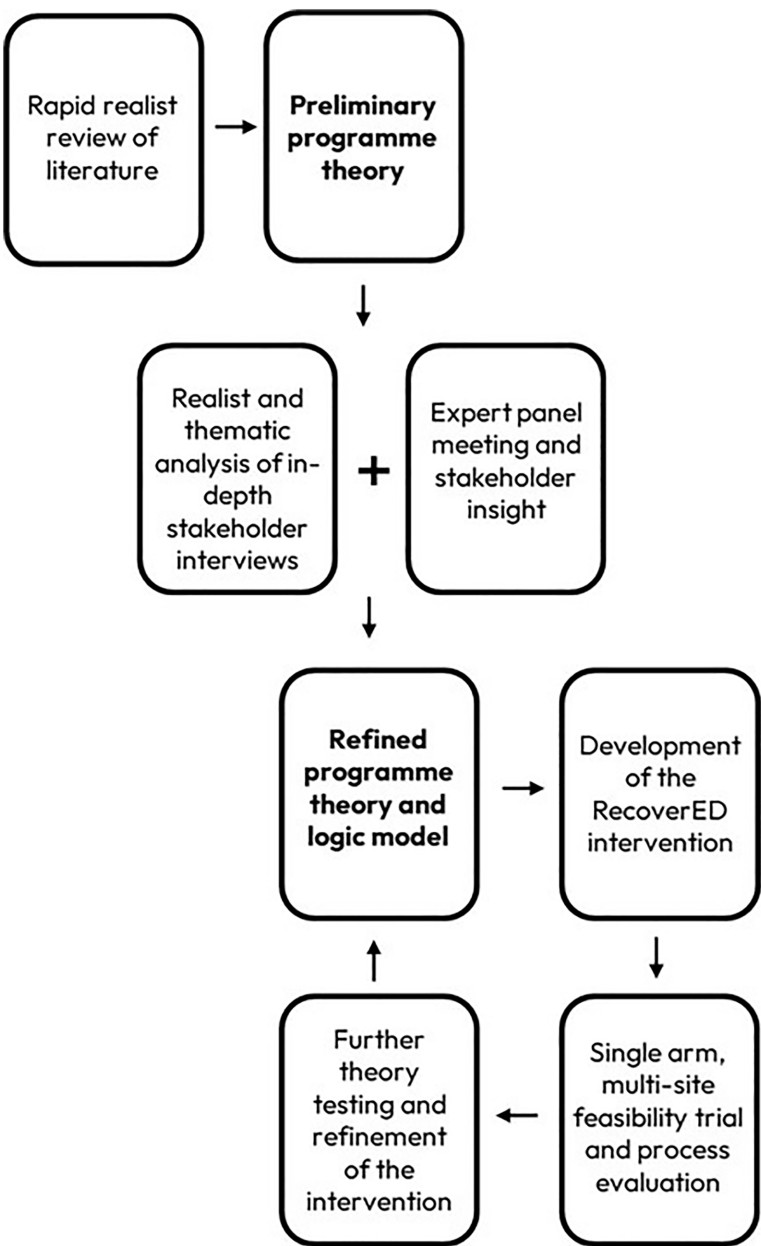

**Fig 1. Realist-informed development process of the RecoverED intervention.**

## Phase 1: Rapid realist review of literature

In the first stage, a rapid realist review of literature was undertaken to identify strategies that might improve recovery from delirium, and how, why, in what circumstances and for whom they should work [11]. A preliminary programme theory was developed following a two-stage synthesis of 52 studies. First, the focus was on studies evaluating the efficacy of interventions to support recovery from delirium, and subsequently, a wider search strategy was used to identify relevant studies on similar patient groups or those using wider methodologies. The data were synthesised using realist principles, and helped identify, articulate, and consolidate explanatory accounts of how particular intervention components could lead

to proposed outcomes. These explanatory accounts served as 'building blocks' for a broader integrated theory on how best to support recovery from delirium at home [12], in order to inform intervention development. A preliminary programme theory consisting of three interdependent recovery domains and four recovery facilitators was identified. This is shown in Fig 2.

## Phase 2: In-depth interviews with stakeholders

Older people with delirium, their carers, and health and social care professionals took part in semi-structured realist interviews between September 2021 and July 2022. Semi-structured topic guides, based on the preliminary programme theory and a prior realist review, were developed for each group aligning with Manzano's [13] 'Phase I. Theory gleaning interviews'.

A total of forty-six participants were interviewed, comprising eight older adults with delirium, fourteen carers, and twenty-four health and social care professionals, all recruited from an NHS hospital in southwest England. Interviews were conducted via telephone in accordance with COVID-19 safety protocols. All participants provided voluntary informed consent prior to participation. Two older adults with delirium and two carers provided verbal consent due to limited access to printing or scanning facilities, while all other participants provided written consent. Realist interviewing techniques were used to elicit insights relevant to theory through exploration of the research question with participants, as well as their views and feedback on the preliminary programme theory [14]. Data were analysed using concurrent realist and thematic approaches. The aim was to generate a deeper understanding of contextual factors (context mapping; [10]), identify unmet rehabilitation and support needs, and align conceptual gaps in community-based delirium care pathways to the context and desired outcomes. This qualitative research is published elsewhere [15].

## Phase 3: Expert panel workshop

Next, a panel of experts consisting of researchers, medical professionals, allied health professionals, and patient and public involvement representatives were invited to critique and offer recommendations on refining the proposed causal pathways of the planned intervention. The online full-day workshop involved presenting findings from each of the previous stages of the realist research to stimulate structured reflection and discussion. Panellists were encouraged to consider the plausibility, coherence, and explanatory power of the proposed pathways, as well as their relevance and feasibility in practice. This process helped to iteratively build and refine a theory of community-based delirium rehabilitation, with

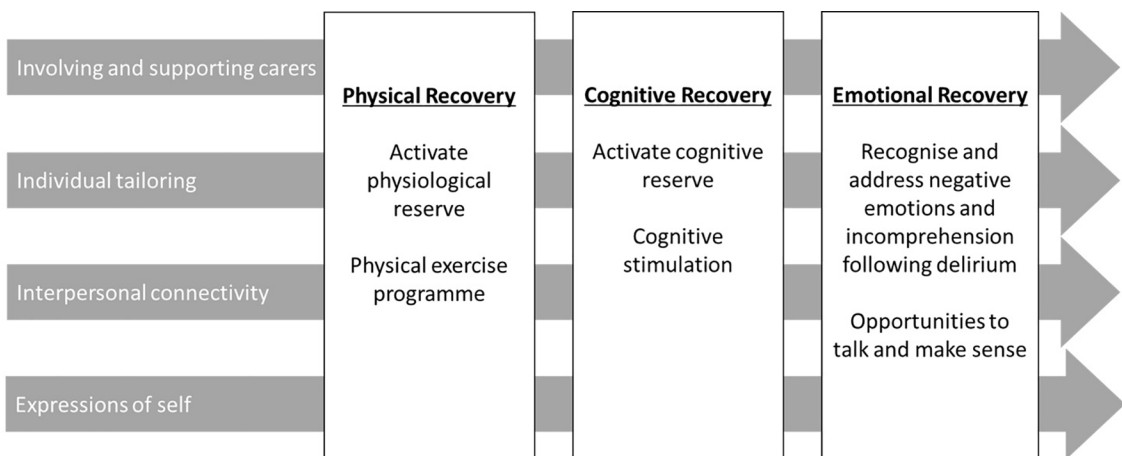

**Fig 2. Preliminary programme theory from O'Rourke et al., (2020).**

particular attention to context mapping, identifying relevant short- and long-term outcomes, and aligning potential intervention components accordingly.

At this stage of the enquiry, context mapping was employed as a structured approach to identifying and theorising the contextual contingencies that influence how mechanisms are triggered, and outcomes are produced, in keeping with Pawson's [16] emphasis on understanding what works, for whom, in what circumstances, and why. Context within healthcare interventions can refer to physical, organisational, sociocultural, political, or economic aspects of the health and care system, policy or public health contexts in which interventions are implemented. It may also be defined as 'social rules, values, sets of interrelationships' that operate within times and spaces that either constrain or support the activation of particular programme mechanisms [14: p. 70].

A broad range of ideas was welcomed to ensure diverse perspectives were considered, and feasibility and practical constraints were used as guiding principles to prioritise and shape the intervention content. The online workshop was audio-recorded and transcribed, and key insights were distilled by the research team to ensure that proposals remained grounded in what could realistically be implemented.

## Phase 4: Programme theory meetings

Between March 2022 and June 2022, a series of 3-weekly meetings were held by the core research team, consisting of a professor of geriatric medicine, a senior academic clinical psychologist, a senior academic physiotherapist, two senior realist researchers, a resident doctor in internal medicine and a research fellow. Although the team represented a broad multidisciplinary perspective, it did not include a researcher with nursing expertise, which was recognised as a limitation in professional representation. The focus was synthesising the findings from the previous stages of research using realist principles. This approach facilitated the identification of clear and evidence-grounded causal pathways, which provided a structured framework for developing a more refined programme theory. The team built on the insights from the expert panel workshop by engaging in a prioritisation process. The aim was to identify key outcomes relevant to the target population (older people with delirium and their carers) and align intervention activities with the context map to facilitate achievement of those outcomes [10]. The programme theory, captured in both the visual logic model and a more detailed text-based document, formed the basis for the prospective development of the multicomponent RecoverED intervention.

## Phase 5: Embedded process evaluation

Programme theories are valuable because they can improve the formulation of evaluation questions, prioritise which outcomes or other data to capture, and contribute to the development of more effective programmes. They should be iteratively tested and refined through the generation and consideration of available evidence [2,17]. In this study, this was undertaken as part of the embedded process evaluation within a single-arm, feasibility trial of the novel RecoverED intervention. A mixed methods approach was used as part of the feasibility trial; qualitative data were collected through semi-structured interviews with patient, carer, and HCP participants. HCP participants included those involved in planning and delivering the *RecoverED* intervention, specifically occupational therapists, physiotherapists, and rehabilitation support workers within the community rehabilitation team.Quantitative data were gathered from case report forms and training logs completed by HCP participants who delivered the intervention within the feasibility study. Data from both qualitative and quantitative sources were collected between June 2023 and July 2024. All participants with capacity provided written informed consent prior to participation. For six older adults with delirium who lacked capacity at the time of recruitment, consent was obtained via a consultee or legal representative in accordance with ethical guidelines. Of these, three participants subsequently regained capacity and provided their own consent, as specified in the study protocol.

Data were triangulated to provide a comprehensive understanding of the intervention, in line with realist principles [18–21]. The process evaluation examined both implementation fidelity and acceptability, while also reflecting on the programme theory to ensure that its underlying inferences and assumptions were supported by the data [8]). The results of

the process evaluation are published elsewhere. Emerging findings from the process evaluation informed the refinement of the detailed causal pathways and the programme theory, enhancing understanding of how the intervention operated within its implementation context. Fig 3 illustrates the iterative process from the perspective of the programme theory evolution within the RecoverED Project.

## Results

Drawing on realist analysis principles, the programme theory was iteratively developed and visually depicted in a logic model to illustrate the connections between intervention inputs, activities, mechanisms, and outcomes, offering a clearer understanding of how change is expected to occur. The resulting intervention developed from this model was a complex, home-based, multicomponent, delirium rehabilitation intervention which is to be planned and delivered by a trained, multidisciplinary community healthcare team to optimise recovery for older individuals in post-acute settings. In the following sections, we explain how the interconnected components of the intervention are expected to achieve both short- and long-term outcomes, while considering the specific problem contexts that they sought to improve.

Phases 1 and 2, comprising the rapid realist review and qualitative analysis, have been published separately [11,15]. The following sections outline how the programme theory was progressively developed through phases 3, 4 and 5.

### Context map (Phase 3 results)

A context map was developed as part of the theory-building process in Phase 3, during the expert panel workshops, and was subsequently elaborated in the subsequent phases. Renger et al. [10] suggest that a context map can support the integration of context into a programme theory, helping evaluators interpret how diverse contextual factors influence the programme's mechanisms of change and, in turn, how these mechanisms affect programme outcomes. In our project, it was used to develop a broad understanding of the need for a multi-component delirium rehabilitation in the community care pathway, by highlighting the various challenges faced by people with delirium and problems with current care at the micro (individual), meso (organisational) and macro (system) levels.

Fig 4 depicts a context map that was elicited through the multi-stage realist-informed research process, and brings together findings from the rapid realist review, qualitative study and the expert panel meeting [10].

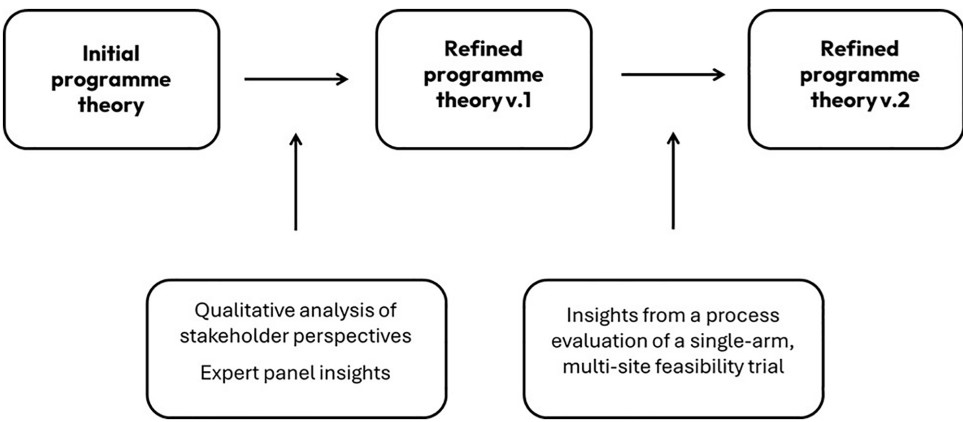

**Fig 3. RecoverED Programme theory development.**

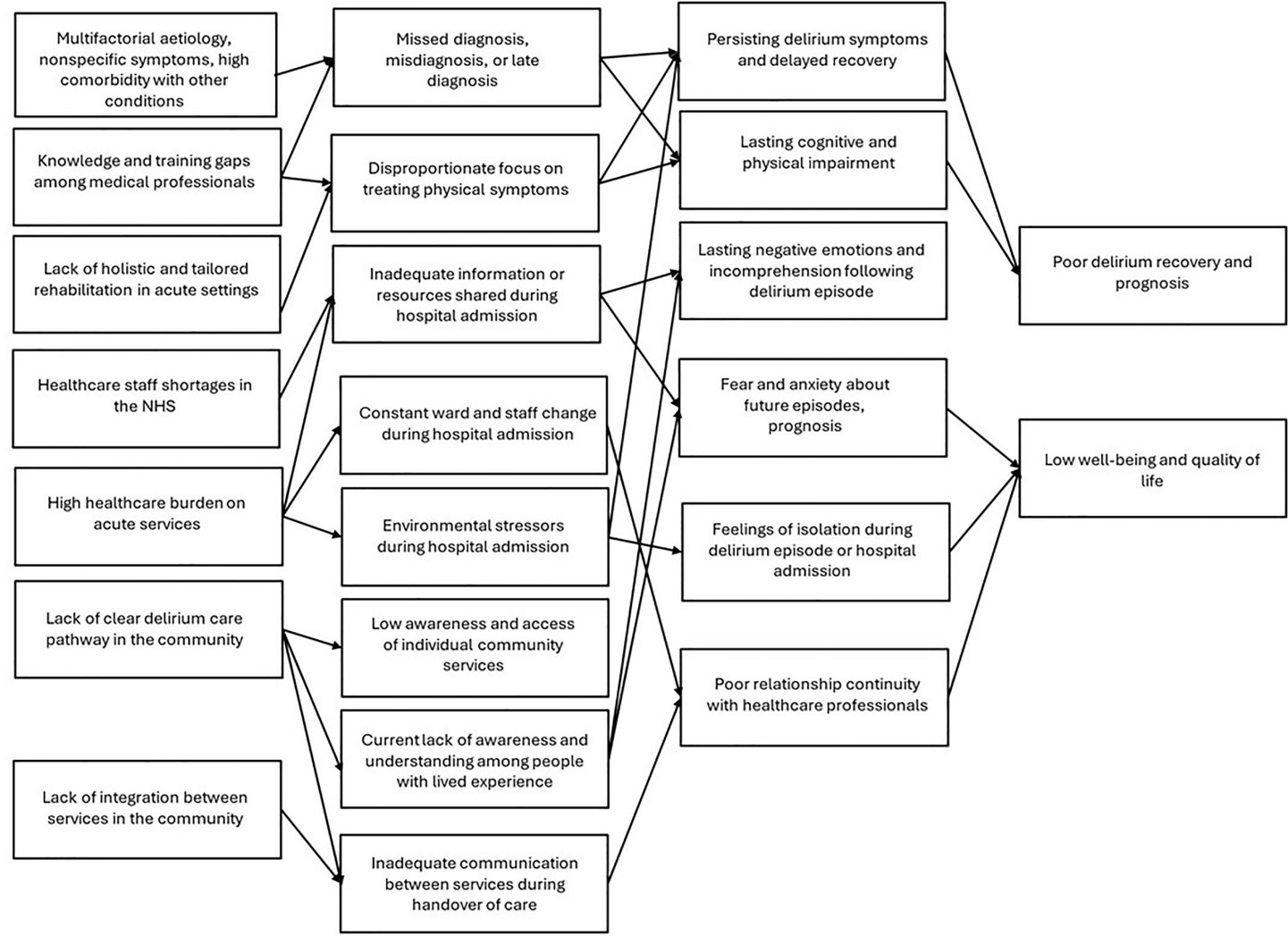

**Fig 4. Context map for community-based impacts and recovery from delirium.**

### RecoverED programme theory

Following the context mapping exercise, the recovery domains and facilitators from the preliminary programme theory [11] were refined into intervention components using emerging evidence from Phase 2 (formative qualitative work) and practical insights from Phase 3 (expert panel meeting). These components were mapped to the context to identify specified and assumed change mechanisms that could improve treatment outcomes for older people with delirium. Refined causal pathways were formulated and developed into a programme theory which integrates internal and external contextual factors that influence implementation, participation, and outcomes [22]. The programme theory v1.0 is presented visually in the logic model, Fig 5 below.

The programme theory begins with an implementation phase, outlining the health system changes required to support delivery of the intervention. The second column specifies the intervention components directed towards older people with delirium and carers, detailing the activities associated with each. The third column presents the hypothesised mechanisms of impact, describing how participants are expected to engage with the resources provided and how these activities are

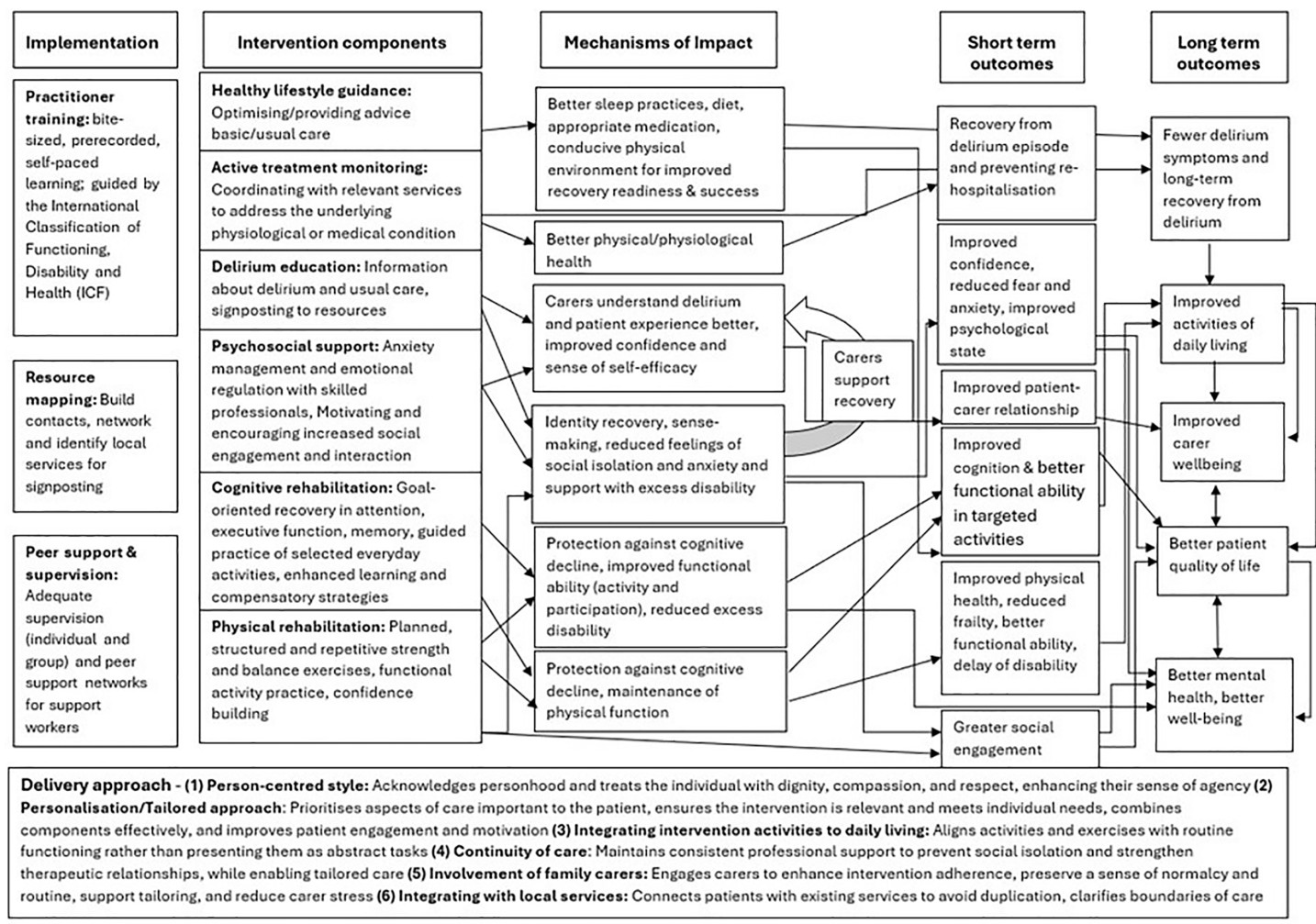

**Fig 5. Logic model v1.0.**

anticipated to generate change. The fourth column sets out the short-term outcomes arising from intervention engagement, followed by the long-term outcomes reflecting the broader, sustained impacts. Finally, the delivery approach highlights recommendations informed by best practice in delirium rehabilitation, with particular emphasis on community-based care.

The sections below demonstrate how each of the components is integrated into the programme theory in a way that clearly describes and delineates the complex relationships between the context, intervention components, change mechanisms and outcomes.

**Implementation phase.** The context map reveals challenges in diagnosing and managing delirium due to its complex presentation, compounded by knowledge and training gaps among healthcare professionals. To address these challenges, the implementation phase includes a comprehensive training programme aligned with the International Classification of Functioning, Disability and Health (ICF) [23]. The training programme covers key topics in delirium rehabilitation within a multicomponent, multidisciplinary model of care. It is structured as bite-sized, pre-recorded modules

to facilitate self-paced learning and is reinforced by peer support and supervision to ensure healthcare professionals delivering the intervention are well supported.

The context map highlighted the absence of a clear delirium care pathway in the community and poor integration between existing services for managing multiple impairments. Community awareness of available services is limited, and the existing services are fragmented, lacking a holistic approach, which complicates access to appropriate support at the right time. To address these challenges, HCPs responsible for planning and delivering the intervention conduct a resource mapping exercise in the implementation phase. This exercise aims to identify existing local services and coordinated support networks to enhance the intervention's delivery while avoiding duplication of services.

**Intervention components, mechanisms of impact, short- and long-term outcomes.** Context mapping indicated that older adults' experiences of delirium within the complex landscape of UK health and social services, across acute and community settings, are often associated with persistent cognitive and physical impairments, psychological distress, and feelings of fear, anxiety, and isolation. The preliminary programme theory identified cognitive, physical, and emotional recovery domains as central to recovery after delirium [11]. Subsequent phases of the realist enquiry provided evidence supporting these key domains, while also highlighting additional rehabilitation strategies that might enhance recovery by activating various mechanisms and responses to facilitate desired recovery outcomes. Context, Mechanism, Outcome configurations accompany the presentation of these components below to illustrate how specific contexts and mechanisms interact to generate outcomes. These components are expected to interact in various ways to produce the desired outcomes and are unlikely to act independently or in isolation to drive change.

*Cognitive rehabilitation*: Delirium can lead to a range of impairments. Drawing on models of disability and rehabilitation, it is hypothesised that addressing cognitive impairments (which may be reversible) and their impact on functional ability, while considering the person's context, will support faster and more complete recovery. To tackle persistent cognitive impairments following a delirium episode, the programme theory suggests personalised, goal-oriented activities to improve attention, executive function, and memory, with progressively increasing challenges to foster sustainable recovery. Additionally, guided practice of everyday activities (using behavioural methods) may enhance task performance and support functional recovery in cognitively complex activities of daily living (ADLs). Improving ADL performance may also involve applying enhanced learning techniques and compensatory strategies, such as aids and environmental adaptations. Table 1 summarises the causal pathways for cognitive rehabilitation.

*Physical rehabilitation*: Delirium is strongly associated with disability, immobility, frailty, increased risk of falls, and overall functional decline. One of the key outcomes for older people with delirium identified in this research is the optimisation of functional capacity, to promote activity and preserve functional reserve and participation. It is hypothesised that physical exercise (i.e., a subset of physical activity that is planned, structured, and repetitive) or physical rehabilitation can help maintain physical health and function, reduce frailty, prevent or delay disability, and improve functional ability of targeted

**Table 1. Cognitive rehabilitation – Causal pathways.**

| Target problem | Intervention actions | Intended outcomes |
|---|---|---|
| Persistent cognitive impairment post-delirium hinders daily functioning and performance of complex ADLs. | Personalised, goal-oriented cognitive activities enhance attention and executive function. | Improved cognitive performance and functional independence. |
| | Guided practice using behavioural methods reinforces learning and task performance. | Increased competence in everyday activities and reduced dependency. |
| Cognitive impairments create uncertainty and lack of confidence in daily tasks. | Compensatory strategies and environmental adaptations support task execution. | Greater self-efficacy, confidence, and engagement in ADLs. |

activities. More indirectly, it is hypothesised that physical rehabilitation can improve functional ability (activity and participation), and reduce excess disability. Table 2 summarises the causal pathways for physical rehabilitation.

*Psychosocial recovery*: Emotional distress, including fear, anxiety, and a sense of threat, is common after delirium, and may be intensified by ongoing or recalled perceptual disturbances and delusions. The emotional impact of delirium can hinder recovery both directly, by making it difficult for individuals to make sense of their experience, and indirectly, by reducing engagement in activities that support recovery, thereby contributing to excess disability. The emotional recovery domain focuses on managing emotional responses and restoring identity. It is hypothesised that anxiety management and behavioural activation (e.g., diary planning) can help regulate emotions, address excess disability, and support cognitive recovery. These strategies can be combined with cognitive and functional tasks to help individuals regain confidence and reduce anxiety.

Delirium is also linked to feelings of isolation, which can be worsened by reduced mobility and cognitive dysfunction, further distancing individuals from social networks. To address this, the emotional recovery domain was expanded to emphasise psychosocial recovery, highlighting the importance of social contact and engagement. Structured activities promoting safe social interactions with support workers or carers can improve social functioning in everyday life, such as socialising, meeting friends, or running errands. This can enhance confidence in performing functional tasks, improve psychological well-being, and ultimately reduce excess disability, supporting recovery from delirium. Table 3 summarises the causal pathways for psychosocial recovery.

*Active health monitoring*: Current delirium care often focuses on identifying and treating underlying medical conditions, such as infections, diabetes, or heart failure, that may trigger or worsen delirium. Poor physiological outcomes can delay hospital discharge, underscoring the need to address physical health as a critical first step in recovery. While insufficient

**Table 2. Physical rehabilitation – Causal pathways.**

| Target problem | Intervention actions | Intended outcomes |
|---|---|---|
| Delirium is linked to disability, frailty, immobility, and increased fall risk. | Structured physical exercise and rehabilitation enhance strength, mobility, and balance. | Improved functional capacity and reduced risk of falls. |
| Older adults with delirium experience functional decline and reduced participation. | Physical rehabilitation maintains physical health and preserves functional reserve. | Greater independence and autonomy, and sustained engagement in daily activities. |
| Frailty and inactivity exacerbate disability and limit recovery. | Planned, structured physical activity prevents disability progression, and enhances activity levels. | Enhanced physical resilience, participation, and prolonged functional ability. |

**Table 3. Psychosocial recovery – Causal pathways.**

| Target problem | Intervention actions | Intended outcomes |
|---|---|---|
| Emotional distress, including fear, anxiety, and a sense of threat, is common after delirium, worsened by perceptual disturbances and delusions. | Anxiety management and behavioural activation (e.g., diary planning) help regulate emotions and provide coping strategies. | Reduced emotional distress, improved psychological well-being, and better engagement in recovery activities. |
| Delirium is associated with isolation, worsened by reduced mobility and cognitive dysfunction. | Encouraging structured social interactions with support workers or carers enhances social engagement. | Improved social functioning, reduced loneliness, and increased motivation for recovery. |
| Limited social contact and engagement negatively impact confidence and functional recovery. | Social activities such as meeting friends or running errands provide reassurance and a sense of normality. | Increased confidence in performing daily tasks, recovery of identity, and reduced excess disability. |

for long-term recovery on its own, treating these underlying conditions improves overall health and supports recovery. Coordinating with relevant services and referring individuals for ongoing care can help manage underlying health concerns and facilitate sustained recovery. Table 4 summarises the causal pathways for active health monitoring.

*Healthy lifestyle practices*: Healthy lifestyle practices that promote and maintain overall health can help prevent future episodes of delirium and reduce the risk of hospitalisation. While treating the underlying cause may support initial recovery from acute delirium symptoms, long-term recovery requires a more holistic approach. Healthcare professionals can assist by guiding individuals on healthy lifestyle changes, such as improving diet, hydration, and sleep routines. Optimising the home environment by removing trip hazards, ensuring sufficient lighting, and maintaining good ventilation is also hypothesised to enhance recovery by creating a safer and more supportive physical context. Table 5 summarises the causal pathways for healthy lifestyle practices.

*Delirium education support*: The context map highlighted low awareness of delirium among those with lived experience and poor communication from hospital staff at discharge, which heightens fear, anxiety, and lack of confidence in managing symptoms. Educational support aimed at the person with delirium and their carer aims to address this by offering opportunities for learning, support, and sense-making with skilled, trained professionals. A greater understanding of delirium can contribute to better symptom management and engagement with rehabilitation interventions. This in turn can lead to greater confidence, a better interpersonal relationship with the carer, and reduced fear and anxiety, resulting in overall improvements in psychological state. Table 6 summarises the causal pathways for delirium education support.

**Intervention delivery approach.** Contextual factors not directly addressed by intervention components or activities are instead managed through a recommended approach to planning and delivering the intervention, rooted in best practices for delirium rehabilitation and care. Our findings highlight the importance of a person-centred approach as central to successful implementation, particularly one that values individuals regardless of cognitive impairment and recognises the role of environmental influences on behaviour [24,25]. It is anticipated that participants will engage

**Table 4. Active health monitoring – Causal pathways.**

| Target problem | Intervention actions | Intended outcomes |
| --- | --- | --- |
| Older people with persistent delirium experience ill health and medical symptoms. | Addressing these conditions stabilises physiological health and prevents further deterioration. | Improved medical stability and readiness for rehabilitation. |
| Poor physiological outcomes delay hospital discharge and hinder recovery. | Timely treatment of medical conditions supports physical recovery and reduces complications. | Faster discharge, reduced hospital stays and improved overall health. |
| Treating underlying conditions supports recovery, but multimorbidity is common and requires holistic management. | Coordinating care and referring individuals to ongoing health services ensure continued management of medical concerns. | Sustained recovery, prevention of relapse, and improved long-term health outcomes. |

**Table 5. Healthy lifestyle practices – Causal pathways.**

| Target problem | Intervention actions | Intended outcomes |
| --- | --- | --- |
| Older people with persistent delirium experience ill health and medical symptoms. | Healthcare professionals provide guidance on diet, hydration, and sleep routines to promote overall health. | Reduced risk of delirium recurrence and hospitalisation, supporting sustained recovery. |
| The home environment may present hazards that increase the risk of falls and complications. | Guidance on creating a safer and more supportive space can create an environment conducive to recovery. | Enhanced safety, reduced risk of injury, and improved overall well-being. |

**Table 6. Delirium education support – Causal pathways.**

| Target problem | Intervention actions | Intended outcomes |
|---|---|---|
| Low awareness of delirium and poor communication at discharge increase fear, anxiety, and lack of confidence in symptom management. | Educational support provides learning opportunities, sense-making, and guidance from trained professionals. | Improved understanding of delirium, leading to better symptom management and engagement in rehabilitation. |
| Uncertainty about delirium can strain interpersonal relationships and contribute to psychological distress. | Increased knowledge and support empower both the person with delirium and their carer. | Greater confidence, improved carer-patient relationships, and reduced fear and anxiety, enhancing psychological well-being. |

more effectively when their personal recovery goals are prioritised, and their sense of agency and identity is respected. Personalisation and goal setting are expected to support sustained engagement, with tailored activities aligned to individual preferences and abilities. Flexibility in strategies, ranging from simple to complex, may help accommodate diverse recovery needs. Incorporating meaningful activities into daily routines, such as gardening or meal preparation, is theorised to increase engagement and reinforce cognitive gains. The involvement of family carers, supported with appropriate information and guidance, is expected to strengthen routines, personalise care, and reduce stress. Maintaining relationship continuity with professional carers may help build trust, mitigate fear and isolation, and enhance emotional well-being and social interaction. Finally, integrating the intervention with local services is likely to prevent duplication, support holistic care, and enable more effective use of resources. Table 7 summarises the causal pathways for the intervention's delivery approach.

## Reflections on the programme theory

Within the process evaluation embedded in the single-arm feasibility trial of the RecoverED intervention, we considered how the initial programme theory (v1.0) was reflected in implementation and participant experiences. While this did not amount to a formal evaluation of the programme theory, the process evaluation generated insights that informed minor refinements. These adjustments did not alter the core assumptions or overall structure of the programme theory. A synthesis of key insights across all data sources is presented below, and minor revisions to the programme theory are shown in Fig 6 (logic model v2.0), with changes denoted using underlining.

**Implementation phase: Insights and modifications.** The implementation phase involved some modifications to the intervention following the process evaluation. Although the training was comprehensive, its length and format may need to be reconsidered to accommodate diverse learning styles. HCPs appeared to favour face-to-face, interactive methods, including practical sessions, case studies, and demonstration videos, rather than prerecorded module-based presentations. The RSW's background and level of experience appeared to influence the support required. Multidisciplinary expertise within the team seemed important for effective planning and supervision. The psychosocial component, in particular, appeared to require additional support and supervision for HCPs, with ad hoc support from the study clinical team being viewed as valuable. Findings also suggested that dedicated recovery teams within the same healthcare organisation may foster trust and facilitate effective teamwork. With regard to resource mapping, local service research appeared to be conducted iteratively, driven by immediate need and local availability rather than through a planned, prospective approach as anticipated in Programme Theory v1.0.

**Insights and modifications: Recovery components.** Participants appeared to show strong support for the causal pathways outlined in Programme Theory v1.0, suggesting these strategies were viewed as relevant for supporting recovery in older people post-delirium. Cognitive rehabilitation was consistently planned around participant-led goals and aligned with routine occupational therapy for older people with delirium. However, one HCP noted that dealing with delirium and its underlying causes is not typically considered part of occupational therapy, highlighting a potential

**Table 7. Intervention delivery approach – Causal pathways.**

| Target problem | Intervention actions | Intended outcomes |
|---|---|---|
| Many people with delirium experience loss of dignity, agency, and misaligned recovery goals due to cognitive impairment and environmental factors. | A person-centred approach that values individuals regardless of cognitive impairment and considers environmental influences. | Participants are treated with dignity, their personal recovery goals are prioritised, and their sense of agency is preserved. |
| Recovery needs vary widely due to differences in symptom presentation, abilities, preferences, and motivation. | Personalisation and goal setting with flexible strategies tailored to individual needs. | Enhanced motivation, sustained engagement, and improved recovery outcomes. |
| Frequent changes in professional carers and fragmented care can lead to confusion, fear, and a sense of isolation among individuals recovering from delirium. | Ensuring continuity of care by maintaining consistent relationships with professional carers builds trust and improves communication. | Enhanced emotional well-being, increased confidence, and improved social engagement, supporting more effective recovery. |
| Interventions that do not incorporate insights from family carers may lack the personal details necessary for tailoring care to the individual's unique needs and preferences. | Involving family carers in the care planning process provides valuable insights into the individual's history, routines, and preferences, enabling a more personalised approach. | The intervention is better tailored to the individual's needs, leading to improved engagement, adherence, and recovery outcomes. |
| Cognitive exercises may seem abstract if not connected to daily life. | Integrating intervention activities into everyday routines (e.g., gardening, meal preparation) to make tasks more relevant. | Increased engagement and more effective functional recovery. |
| Disconnected local services can lead to duplication and inefficient resource use. | Integrating the intervention with local services to ensure holistic care coordination. | Optimised resource use, cost-effective delivery, and comprehensive support. |

need for change. While some participants and carers found cognitive rehabilitation challenging, many responded positively and showed cognitive improvements, increased self-reported competence in daily activities, and better functional independence. Though it was unclear whether self-efficacy improved, participants appeared to demonstrate higher confidence in independent functioning. Those with severe cognitive impairments were less engaged, with some withdrawing due to care home placement.

Physical rehabilitation, including mobility and outdoor exercises, was planned more often than delivered due to external factors such as unsuitable home environments, poor health, or weather conditions. Fear of falls deterred some participants, while others embraced challenges and progressed. HCPs appeared confident in delivering physical rehabilitation, which was often practised between sessions and supported by clear manual guidance. Some participants achieved challenging goals, with observed and self-reported improvements in functional capacity, independence, and engagement in daily tasks. Long-term effects on fall risk remain unclear. Integration with community physiotherapy was considered beneficial for those with additional needs, and tailoring interventions to individual abilities was regarded as essential and useful. No significant changes to the causal pathways for cognitive or physical rehabilitation were identified.

The psychosocial component appeared to be the least familiar aspect for health and social care professionals, requiring the most guidance for planning and delivery. This was further complicated for some participants who had complex mental health needs beyond the scope of RecoverED. Initially, psychosocial support appeared under-planned, as HCPs were unfamiliar with this type of support. However, as the sessions progressed, the important role of psychosocial support became clearer to HCPs, leading to its increased delivery. This change suggested that HCPs' practice may adapt in response to emerging insights and supported the theory on the central role of psychosocial support in delirium

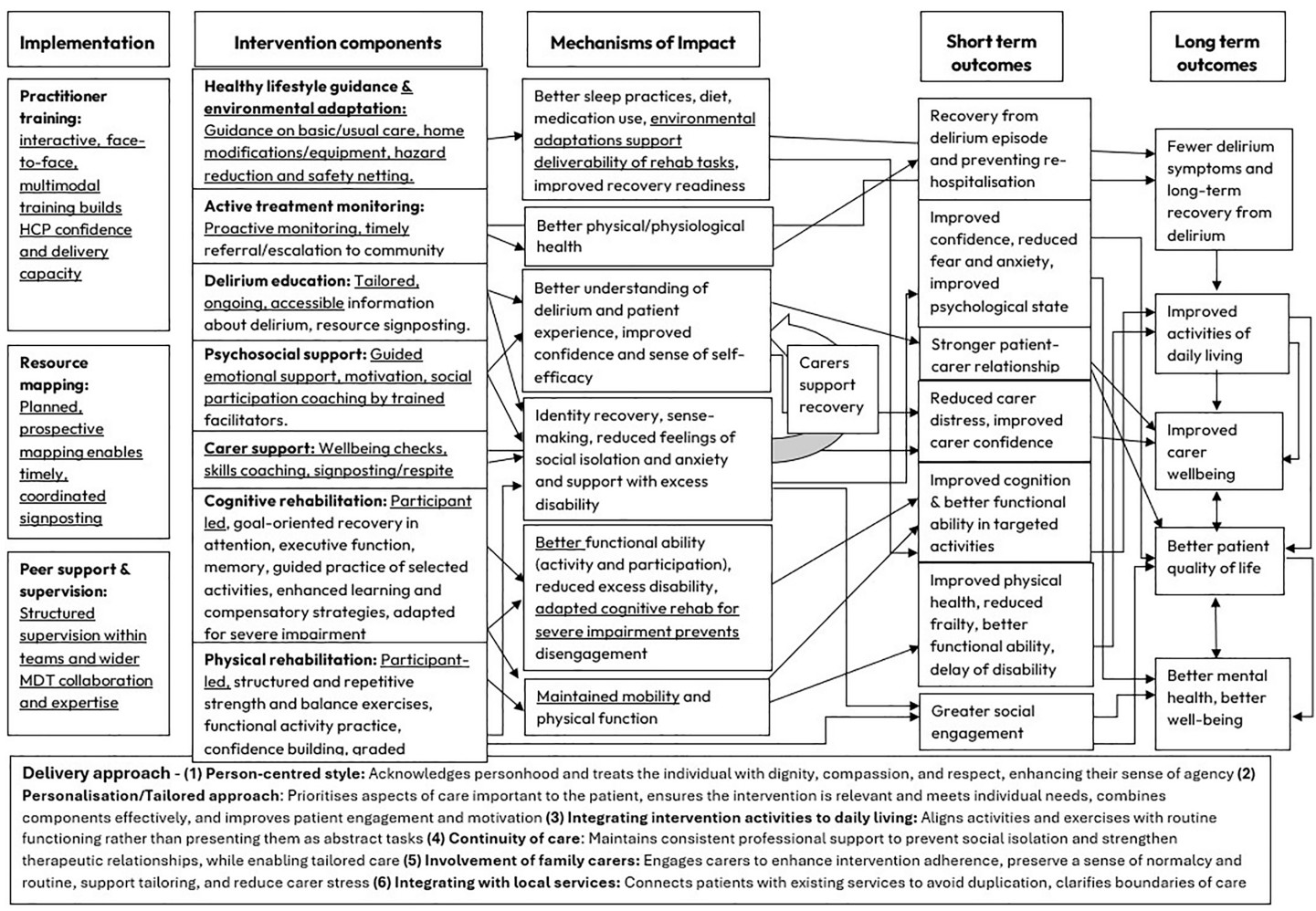

**Fig 6. Logic model v2.0.**

rehabilitation. At the same time, some participants were reluctant to discuss their emotional well-being, highlighting the importance of tailoring activities to individual preferences and readiness to engage.

Many participants expressed a desire to socialise with family, friends, and neighbours, and the interpersonal nature of the intervention was widely valued. Participants appeared to appreciate the social interaction with HCPs and often built positive relationships with the team. Carers experienced significant distress but sometimes struggled to seek help for their own well-being. While carer support is included in the programme theory and intervention design, findings highlighted a potential need to strengthen this component, as some carers hesitated to ask for help. Engaging in psychosocial support appeared to help reduce psychological distress and resolve negative emotions related to past experiences of delirium for some participants and carers, leading to reported improvements in subjective wellbeing. It was unclear whether HCPs actively planned and delivered activities to build confidence in social engagement, but most respondents agreed it was essential for recovery. While no major changes to the causal pathways were identified, findings pointed to a need for greater personalisation of psychosocial support and stronger emphasis on carer support within the theory.

Active health monitoring and healthy lifestyle guidance were often planned and delivered within the programme. HCPs frequently signposted participants to community services, such as GPs, rapid response teams, and carer support

services, particularly when participants had not been assessed by other NHS services post-discharge. Participants responded differently to healthy lifestyle guidance; while some required support, others were already well-informed. Advice on sleep, diet, nutrition, and hydration was commonly provided during sessions. Carer involvement appeared crucial in implementing healthy lifestyle practices, as illustrated by one participant living alone who deteriorated due to poor nutrition and hydration. The home environment also influenced rehabilitation, with some homes being unsuitable for practising interventions, necessitating adaptations and safety measures. Observations highlighted the importance of active health monitoring, timely medical treatment, and healthy lifestyle practices in supporting rehabilitation. No changes were suggested to the causal pathways for these components.

Delirium education was provided via a leaflet at the start of the intervention. Participants valued the leaflet, finding the format and language clear and accessible. However, most did not use it beyond an initial read and did not seek further clarification or support from the RSW during the intervention. While information support immediately post-discharge was valued, the optimal format for ongoing education remains unclear. Some participants were unable to engage with the resource due to cognitive impairment, suggesting that in some cases carers may be the more suitable audience. Given the evidence supporting the need for consistent delirium education and awareness, this component may require revision in both format and delivery. It remains uncertain whether delirium education directly improved symptom management, rehabilitation engagement, participant–carer relationships, or overall psychological well-being. No changes to the causal pathways were suggested, but the format of implementation may need to be refined and evaluated in future stages.

**Insights and modifications: Delivery approach.** There was consistent indication that HCP teams adhered to the intervention delivery approach, resulting in good uptake, perceived value, and overall satisfaction. Participant-led goals appeared to ensure that patient agency and autonomy were prioritised over generic rehabilitation outcomes, with activities often targeting improvements in functional tasks relevant to individual daily routines and goals. Nearly all participants felt the intervention was personalised to their needs and abilities.

Carers attended most sessions and were either actively involved or given the option not to participate, based on their preferences. Lack of involvement was typically due to confidence in the RSW's abilities and a perceived lack of need for their participation in certain activities, such as mobility exercises or discussions about emotional topics.

Continuity of care appeared to be an important factor in perceived success, as all participants worked with the same HCP team throughout the intervention, with no staff attrition. This continuity seemed to foster trusting relationships between HCP teams and participants, enhancing engagement and satisfaction. Rehabilitation was also integrated with existing community services based on individual needs.

**Contextual considerations.** Useful insights into the context map were generated through this process. Healthcare teams are often understaffed and face competing demands, making complex interventions like RecoverED challenging to implement. Successful delivery appeared to require strong MDT collaboration, clear planning, and structured debriefing, supported by comprehensive training. This training may need to go beyond written materials to include case studies, demo videos, and interactive learning strategies to reflect the complexity of the content.

To support self-sufficiency and effective problem-solving, supervision systems within teams may need to be strengthened, with a focus on maintaining multidisciplinary expertise. Geographical proximity, both within teams and to assigned participants, appeared important for overcoming practical challenges in intervention planning, assessments, and supervision.

Carers seemed to play a pivotal role in the uptake and success of rehabilitation, yet their own support needs may remain insufficiently addressed. Psychosocial support also appeared to be undervalued, and stronger community-based signposting may be required to assist older adults with delirium, particularly those with additional complex needs that extend beyond the scope of rehabilitation interventions.

## Discussion

This paper outlines the iterative development and evaluation of the programme theory for a home-based rehabilitation intervention aimed at improving recovery after delirium, drawing on realist approaches. By drawing from multiple data sources – including a rapid realist review, thematic analysis of data from participants and healthcare professionals, expert panel consultations, and a process evaluation embedded within a feasibility trial – we constructed a theoretically grounded framework that provides insights into what works, for whom, and in what contexts. This ensured that the theoretical foundation was robust, contextually grounded, and informed by the latest research and practice.

A multistage, multimethod approach was central to the realist theory-building process, reflecting the complex and context-dependent nature of the intervention. This approach enabled the iterative refinement of our programme theory. Engaging multiple stakeholders, including participants, practitioners, and experts, ensured that the theory was contextually relevant, reflective of diverse perspectives, and practically applicable. This comprehensive approach strengthens the validity and utility of the framework, providing a deeper understanding of how the intervention operates across different settings and populations.

Using propositions about causal pathways, we identified how specific intervention actions, were expected to contribute to intended outcomes. Realist processes typically reveal multiple interacting contexts, mechanisms and outcomes, shaped by individual, institutional and infrastructural factors, resulting in varied patterns of success and failure (22; p. 185). In our case, mapping out the different elements of the intervention highlighted multiple plausible theories of change based on overlapping components and activities, each offering a distinct explanation of how the intervention works in context to achieve its goals.

These components are expected to interact differently depending on individual symptom profiles, personal preferences, and environmental conditions. This reflects current thinking in non-pharmacological delirium treatment, where flexible, multi-domain interventions are seen as most effective [26]. As Mukumbang et al., [27] also observed, a range of contextual factors shaped how mechanisms activated responses. Engagement varied, with cognitive and physical rehabilitation activities generally well-received, while psychosocial and lifestyle components were met with more mixed responses. This does not alter our theoretical propositions but underscores the need to address participant perceptions and behaviours to sustain engagement. It also highlights the importance of a personalised approach to rehabilitation, with adaptive delivery models that align with individual readiness and preferences.

This study contributes to the growing literature supporting realist principles in health services research. Although their benefits in this field remain debated [2,28], our findings show that a programme theory-informed approach can effectively unpack complex, community-based interventions, providing a nuanced, theory-driven understanding of mechanisms and outcomes [29]. By organising the programme theory into interrelated causal pathways, we emphasise links between components and outcomes, offering a framework for future research facing similar methodological challenges [21]. Calls for stronger theory-driven approaches in healthcare research [2] highlight the importance of flexible, adaptive frameworks. The RecoverED intervention, developed through a realist-informed, iterative, multi-stage process, was well-aligned with implementation contexts, and the process evaluation required only minor refinements, demonstrating the robustness of this development approach.

Our findings suggest that carer involvement is likely important for intervention uptake and patient recovery, though this is inferred from the programme theory rather than empirically demonstrated. Support structures for carers are currently limited within both the theoretical framework and implementation strategy, highlighting the need for research on how variations in carer involvement affect adherence and outcomes. Without adequate support, carers often experience emotional distress due to caregiving demands and uncertainty around delirium recovery [30] which can compromise their ability to provide effective care. Rehabilitation plans should therefore incorporate structured support mechanisms, such as formal training, peer networks, and access to professional mental health resources, benefiting carers and enhancing patient recovery. While the involvement of carers was central to the *RecoverED* approach, we also recognise that some

individuals experience delirium without access to a carer. Identifying ways to extend support to those living independently will be an important focus for future development. This study also identified operational challenges, such as service integration and resource availability, which must be addressed before wider implementation. Strengthening partnerships with community services, refining referral pathways, and embedding the intervention within routine care are essential for scalability. Additionally, a more structured delirium education component is needed to ensure long-term engagement with rehabilitation efforts.

## Limitations

A key limitation of our work is that outcome data were not incorporated into the assessment and evaluation of the programme theory. Although we gained valuable insights into implementation, contextual influences, and perceived mechanisms of engagement, we did not directly assess the intervention's effectiveness in achieving outcomes within the theory evaluation. Our findings focus on explaining how and why the intervention is believed to work, rather than its rigorously assessed impact. This is reflected in the revised logic model, which has not undergone substantial changes to the underlying theory at this stage. We intend to refine and adapt the theory more fully in future, when undertaking a comprehensive realist evaluation of the programme. Another limitation is the concentration of the qualitative research in Phase 2 within a single geographic area (southwest England), which may limit the transferability of the findings to other settings. Since context plays a vital role in shaping how mechanisms operate, the findings from a single setting may not fully apply to other contexts. Future research should explore how similar interventions function in diverse geographic, socio-demographic, ethnic patient populations and healthcare settings to assess the generalisability of the programme theory.

## Conclusion

This study contributes to the growing field of realist-informed intervention research by detailing the development and refinement of a programme theory for home-based delirium rehabilitation. Using the RecoverED intervention as a case study, we demonstrated how developing an explicit programme theory, as advocated by the MRC guidance on complex interventions, can make transparent the hypothesised links between intervention components, mechanisms of action, contextual influences and outcomes. In developing the programme theory, we identified plausible and testable causal pathways linking the intervention to improvements in cognitive, physical and psychosocial recovery, while highlighting the critical role of carer involvement and context-sensitive implementation.

Few studies have applied theory-informed evaluation in this area, and this study offers an empirically grounded framework to guide future research and practice, while also illustrating the value of flexible, theory-driven approaches. The next phase of our research will test and refine these theories through a controlled trial, validating the mechanisms of change and assessing the programme's wider effectiveness.

## Supporting information

**S1 File. Anonymised data - repository details.** Document containing the repository details and permanent link to the anonymised minimal dataset underlying the results reported in this study.
(DOCX)

## Acknowledgments

The authors sincerely thank the older adults, family carers, and health and social care professionals who generously shared their time and insights throughout the iterative, multi-phase development of the RecoverED intervention. We are grateful to the NHS Trusts that supported participant recruitment and data collection during the feasibility trial and embedded process evaluation. We also thank the stakeholders whose input during the expert panel workshop and programme

theory meetings was invaluable in refining the intervention and its underlying programme theory. Our appreciation extends to the wider research team for their contributions to synthesising findings, integrating insights, and developing the RecoverED intervention.

## Author contributions

**Conceptualization:** Shruti Raghuraman, Sarah Morgan-Trimmer, Rob Anderson, Linda Clare, Rowan Harwood, Louise Allan.

**Data curation:** Shruti Raghuraman, Sarah Morgan-Trimmer, Rob Anderson, Linda Clare, Ellen Richards, Aseel Mahmoud, Louise Allan.

**Formal analysis:** Shruti Raghuraman, Sarah Morgan-Trimmer, Rob Anderson, Victoria A Goodwin, Linda Clare, Ellen Richards, Aseel Mahmoud, Louise Allan.

**Funding acquisition:** Linda Clare, Louise Allan.

**Investigation:** Shruti Raghuraman, Aseel Mahmoud, Alison Bingham, Louise Allan.

**Methodology:** Shruti Raghuraman, Sarah Morgan-Trimmer, Rob Anderson, Victoria A Goodwin, Linda Clare, Rowan Harwood, Louise Allan.

**Project administration:** Jinpil Um, Louise Allan.

**Supervision:** Sarah Morgan-Trimmer, Rob Anderson, Victoria A Goodwin, Linda Clare.

**Writing – original draft:** Shruti Raghuraman.

**Writing – review & editing:** Sarah Morgan-Trimmer, Rob Anderson, Victoria A Goodwin, Linda Clare, Ellen Richards, Aseel Mahmoud, Alison Bingham, Elizabeth Goodwin, Rowan Harwood, Annie Hawton, Sarah Joanna Richardson, Jinpil Um, Obioha C Ukoumunne, Louise Allan.

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
