## [Decision Letter · Decision Letter 0]

28 Oct 2025

Dear Dr. Raghuraman,

Thank you for submitting your manuscript to PLOS ONE. After careful consideration, we feel that it has merit but does not fully meet PLOS ONE’s publication criteria as it currently stands. Therefore, we invite you to submit a revised version of the manuscript that addresses the points raised during the review process.

We look forward to receiving your revised manuscript.

Kind regards,

Tim Luckett

Academic Editor

PLOS ONE

Journal Requirements:

2. Please describe in your methods section how capacity to provide consent was determined for the participants in this study. Please also state whether your ethics committee or IRB approved this consent procedure. If you did not assess capacity to consent please briefly outline why this was not necessary in this case.

3. In the ethics statement in the Methods, you have specified that verbal consent was obtained. Please provide additional details regarding how this consent was documented and witnessed, and state whether this was approved by the IRB.

5. We noted in your submission details that a portion of your manuscript may have been presented or published elsewhere. Figure 2. Preliminary programme theory

Adapted from O’Rourke et al., 2020 (PMID: 32734773), representing Stage 1 of programme theory development. This initial framework was subsequently refined and expanded through the multi-stage process reported in the present study.

6. We note that you have indicated that there are restrictions to data sharing for this study. For studies involving human research participant data or other sensitive data, we encourage authors to share de-identified or anonymized data. However, when data cannot be publicly shared for ethical reasons, we allow authors to make their data sets available upon request. For information on unacceptable data access restrictions, please see http://journals.plos.org/plosone/s/data-availability#loc-unacceptable-data-access-restrictions.

7. In the online submission form, you indicated that data are available upon reasonable request.

8. We note that the grant information you provided in the ‘Funding Information’ and ‘Financial Disclosure’ sections do not match.

9. Thank you for stating the following financial disclosure:

This study is funded by a start-up grant from the University of Exeter and by the National Institute of Health and Care Research Programme Grants for Applied Research Programme (NIHR202338).

Professor Goodwin is a National Institute for Health and Care Research (NIHR) Senior Investigator. The views expressed in this article are those of the author(s) and not necessarily those of the NIHR, or the Department of Health and Social Care.

10. We note you have included a table to which you do not refer in the text of your manuscript. Please ensure that you refer to Table 1, 2, 3, 4, 5, 6 and 7 in your text; if accepted, production will need this reference to link the reader to the Table.

Reviewers' comments:

Reviewer's Responses to Questions

**Comments to the Author**

1. Is the manuscript technically sound, and do the data support the conclusions?

Reviewer #1: Yes

Reviewer #2: Yes

2. Has the statistical analysis been performed appropriately and rigorously?

Reviewer #1: N/A

Reviewer #2: N/A

3. Have the authors made all data underlying the findings in their manuscript fully available?

Reviewer #1: Yes

Reviewer #2: No

4. Is the manuscript presented in an intelligible fashion and written in standard English?

Reviewer #1: Yes

Reviewer #2: Yes

Reviewer #1: Thank you for submitting your manuscript entitled ‘Developing a programme theory of a complex, home-based rehabilitation intervention for recovery after an episode of delirium’.

Your work is extremely important, and I read your manuscript with great interest. I really enjoyed the research and program journey that this manuscript details and explores. Your manuscripts provide valuable information to assist with knowledge translation and replication.

It is fantastic to see delirium recovery models of care being developed. I look forward to seeing further publications about RecoverED.

Overall comments:

Abstract is clear and succinct.

Introduction is also clear and succinct.

Your methods are well described. Figure 1 is great.

Phase 4: the research team consisted of a wide MDT; however, the voice of nursing was missing. Can you please clarify this?

Page 9: Who were the HCP participants? Suggest making this clearer. I think it is important to show that you captured all the voices of the MDT, including nursing. If you did not capture the voices of nursing staff, then I think this needs to be discussed as a limitation.

Results were clear and easy to follow. I think Figure 4 is brilliant. Figures 5 & 6 were both easy to follow and compliment the text.

I wonder if there were any negative attitudes towards delirium that need to be challenged/ addressed in order to make the program success? Did the education challenge attitudes?

I really enjoyed reading your discussion. I wonder how we can support people who do not have a carer?

Excellent conclusion.

Minor suggestions:

On page 7, please correct the error of the bracket in this sentence: ‘The aim was to generate a deeper understanding of contextual factors (context mapping; (10), identify unmet rehabilitation and support needs, and align conceptual gaps in community-based delirium care pathways to the context and desired outcomes.’ I think there is a bracket missing after ‘context mapping’?

Page 10: referencing error (Wong 2015).

Page 10: Can you please provide a reference for the process evaluation published elsewhere?

Figure 6: I am glad you changed to face-to-face interactive learning.

Reviewer #2: This paper has interesting and comprehensive methods and is well written and clearly reported. There is a clear description of who was involved in each stage. Interesting logic model of how the intervention is presumed to work produced.

I note that data will be made available upon request, rather than included in the manuscript.

**Do you want your identity to be public for this peer review?** For information about this choice, including consent withdrawal, please see our For information about this choice, including consent withdrawal, please see our Privacy Policy .

Reviewer #1: No

Reviewer #2: No

---

## [Author Response · Author response to Decision Letter 1]

5 Jan 2026

This has been uploaded as a seperate .docx file.

---

## [Decision Letter · Decision Letter 1]

19 Jan 2026

Developing a programme theory of a complex, home-based rehabilitation intervention for recovery after an episode of delirium

PONE-D-25-52519R1

Dear Dr. Raghuraman,

We’re pleased to inform you that your manuscript has been judged scientifically suitable for publication and will be formally accepted for publication once it meets all outstanding technical requirements.

Kind regards,

Tim Luckett

Academic Editor

PLOS One

Additional Editor Comments:

At the authors' discretion, I wonder whether it's more grammatically correct for the title to read "programme theory FOR" rather than "programme theory OF"? I note that, in the text, the authors refer to "programme theory underpinning" which would be another option to consider for the title.

**Comments to the authors:**

Reviewer #1: All comments have been addressed

2. Is the manuscript technically sound, and do the data support the conclusions?

Reviewer #1: Yes

3. Has the statistical analysis been performed appropriately and rigorously?

Reviewer #1: N/A

4. Have the authors made all data underlying the findings in their manuscript fully available?

Reviewer #1: Yes

5. Is the manuscript presented in an intelligible fashion and written in standard English?

Reviewer #1: Yes

Reviewer #1: Thank you for taking the time to review your manuscript and for addressing my comments.

I have no further comments or concerns.

**Do you want your identity to be public for this peer review?** For information about this choice, including consent withdrawal, please see our For information about this choice, including consent withdrawal, please see our Privacy Policy .

Reviewer #1: **Yes:** Amy MontgomeryAmy Montgomery

---

## [Editor Report · Acceptance letter]

PONE-D-25-52519R1

PLOS One

Dear Dr. Raghuraman,

I'm pleased to inform you that your manuscript has been deemed suitable for publication in PLOS One. Congratulations! Your manuscript is now being handed over to our production team.

Kind regards,

on behalf of

Dr. Tim Luckett

Academic Editor

PLOS One